# Accurate Multiplex qPCR Detection of Epstein–Barr Virus/Cytomegalovirus/BK Virus in Kidney Transplant Patients: Pilot Study

**DOI:** 10.3390/ijms252312698

**Published:** 2024-11-26

**Authors:** Costin Damian, Ramona Gabriela Ursu, Adrian Constantin Covic, Aida Corina Bădescu, Simona Mihaela Hogaș, Elena Roxana Buzilă, Alexandru Duhaniuc, Luminița Smaranda Iancu

**Affiliations:** 1Department of Preventive Medicine and Interdisciplinarity (IX)—Microbiology, Faculty of Medicine, “Grigore T. Popa” University of Medicine and Pharmacy, 700115 Iasi, Romania; costin-damian@umfiasi.ro (C.D.); ramona.ursu@umfiasi.ro (R.G.U.); luminita.iancu@umfiasi.ro (L.S.I.); 2Nephrology Department, Dialysis and Renal Transplant Center, “Dr. C.I. Parhon” University Hospital, 700503 Iasi, Romania; adrian.covic@umfiasi.ro (A.C.C.); simona.hogas@umfiasi.ro (S.M.H.); 3Faculty of Medicine, “Grigore T. Popa” University of Medicine and Pharmacy, 700115 Iasi, Romania

**Keywords:** kidney transplant, graft rejection, EBV, CMV, BKV, PCR

## Abstract

Chronic kidney disease is a really important heath issue, and transplantation is an intervention that can greatly increase patient quality of life and survival. The aim of this study was to perform a comprehensive evaluation of the BK virus, CMV, and EBV in kidney transplant recipients (KTRs); to assess the prevalence of infections; and to test if our detection method would be feasible for use in follow-ups with KTRs. A total of 157 KTRs registered at the Clinical Hospital “Dr. C. I. Parhon”, Iași, Romania, were selected using specific inclusion/exclusion criteria. We tested the blood samples from each patient for BK, EBV, and CMV using a multiplex real-time PCR (qPCR) assay and the TaqMan PCR principle. The highest prevalence was detected for BKV (11/157, 7%), followed by CMV (9/157, 5.7%) and EBV (5/157, 3.2%). By simultaneously detecting three possible nephropathic viruses and oncogenes in KTRs using multiplex real-time PCR, we aimed to optimize their monitoring and follow-up. The prevalence of the tested nephropathogenic viruses—BKV, CMV, and EBV—was comparable to that analyzed in other studies. We demonstrate that the use of qPCR for viral detection in KTRs is a robust, cost-effective method for case monitoring.

## 1. Introduction

Impaired kidney graft function occurs as an early complication after kidney transplantation. Over the last ten years, the number of scientific papers published on this condition has been increasing (more than 2000 papers have been published in the last 10 years). The majority of papers have been published in the USA, mainly at the University of California [1]. The International Society of Nephrology (ISN) also published the Global Kidney Health Atlas—an update on the differences in the global burden and care of patients with kidney disease in different countries and regions of the world—in *The Lancet Global Health* in March 2024. The ISN intended to determine the status and capacity of countries to offer access to adequate kidney care, particularly in areas with low economic resources. Data from 161 countries were stratified based on the ISN geographical regions and World Bank income groups. Our attention was drawn particularly to data from our area: Eastern and Central Europe have the highest prevalence (12.8%) compared to the global median prevalence of chronic kidney disease (9.5%). In total, 74% of the 164 countries were able to provide renal replacement therapy (RRT) to more than half of people with kidney failure. This therapy had average annual per-person costs of around USD 19,400 for hemodialysis, USD 19,000 for peritoneal dialysis, and USD 26,900 for the first year of renal transplantation. The overall median prevalence of nephrologists was 11.8 per million, with significant differences between low-income and high-income countries, representing an 80-fold difference. These numerical data affirm the importance of quantifying and presenting the global burden of kidney disease, and the fact that low-resource countries have a greatly reduced capacity to provide nephrology care [2].

Kidney transplantation is a recommended therapeutic intervention for patients with end-stage renal disease, offering longer survival compared to alternative therapies. According to a study conducted in the Netherlands in 2019, elderly individuals represent an important group within the transplanted population: 30% of new kidney transplant recipients were 65 years of age or older in 2019; in comparison, in 2005, this population group represented only 15% of recipients [3]. The majority of research indicates that older transplant recipients have a survival advantage compared to patients on a waiting list, with only a small number of studies not identifying any survival advantage. However, older kidney transplant recipients have up to a three-fold increased risk of mortality in the first 3–12 months after transplantation. Kidney transplantation is also linked to an improvement in health-related quality of life, which may be a significant factor when considering a kidney transplant in this group. This is supported by a study published by Boer et al that compared the data on quality of life in 115 patients 65 years of age and over on the kidney transplant waiting list with those of 115 transplant recipients in the same age range one year after surgery. Health-related quality of life, both emotional and physical, was higher in the post-transplant population than in patients on the waiting list. For the 46 patients with data available both before and after transplantation, paired analyses showed a trend toward higher emotional quality of life and significantly higher physical quality of life for transplant recipients. Of all variables assessed, patient-reported immunosuppressive drug-related side effects had the highest significant negative correlation with both mental and physical health-related quality of life [4].

A recent study from March 2024 mentioned the greater economic burden and the increased impacts on health-related quality of life in kidney transplant recipients due to antibody-mediated rejection (ABMR) [5], and another French retrospective multicenter cohort study that analyzed the national transplant database found that 86 patients who received kidney transplants for amyloid A (AA) amyloidosis experienced more favorable rates of survival and lower recurrence rates than previously reported. This study supports the practice of kidney transplantation for patients with AA amyloidosis (a severe and rare disease) who experience kidney failure [6].

Despite advances in surgical procedures, immunosuppressive medication, and patient care, kidney transplant rejection continues to be a major problem affecting the long-term survival of the transplanted organ. Although efforts are being made to optimize therapy for rejection episodes, there have been cases of unsuccessful therapy that directly affected graft survival. Even with maximal therapy, some renal allografts may not recover functionally. In addition, acute rejection events may have a detrimental effect on the long-term survival of the transplanted organ despite recovery of function. Renal transplant rejection can present as subclinical, symptomatic, or severe. Typically, subclinical rejection is detected via biopsy after observing an increase in monitoring biomarkers such as donor cell-free DNA or the identification of new donor-specific antibodies (DSAs). Surveillance protocols to identify subclinical rejection differ from center to center, with some relying solely on standard laboratory tests, while others use a combination of DSA assessment, the quantification of other biomarkers, and control biopsies, and there is still no consensus on the optimal method to monitor a potential kidney graft rejection [7].

Borriello et al. noted in a 2022 review that BK virus-associated nephropathy affects up to 10% of kidney transplant recipients and leads to graft loss, with up to 50% of those infected potentially affected. Despite this risk, there are no effective prophylactic measures, and therapeutic options are also limited. In this context, immunosuppressive therapy, which increases the risk of immune rejection, is a risk factor. Based on this rationale, closer monitoring of cases is needed, and the use of rapid, accessible, and easy-to-perform diagnostic methods would lead to increasingly better results [8].

A meta-analysis conducted by a team from the Division of Nephrology and Transplantation in Rotterdam, the Netherlands, found that there is currently no biomarker available other than viral load for monitoring BKV. That study also pointed out that the treatment for BKV nephropathy involves reducing immunosuppressive therapy, which in turn, as previously mentioned, increases the risk of kidney transplant rejection. The solution was found by researchers who discovered that the ELISPOT assay (a highly sensitive method of quantifying cytokine-producing cells) becomes a valuable tool for assessing the risk of disease after it is stimulated by a BKV antigen. When used in conjunction with BKV viremia, it can assist clinicians in determining the appropriate immunosuppressive regimen for patients with active BKV replication [9].

Dakroub et al., a group of researchers from France, found that the serologic status of BKV-infected patients, both donors and recipients, before transplantation can serve as a predictive marker for BKV viral replication after transplantation. The use of different laboratory testing techniques to assess BKV serologic status complicates data processing and makes it difficult to compare test results because of the different cut-off values used to determine seropositivity. Since different results were obtained using the same test, the authors also emphasize the challenges encountered in implementing a serologic diagnostic algorithm for BKV infection that is widely accepted in the clinical setting (e.g., the absence of guidelines in this area leads to different surveillance and therapeutic behaviors for similar cases) [10].

Cytomegalovirus (CMV) infection and disease represent the most prevalent viral infections that occur after renal transplantation. The literature data show that the rate of CMV infection and illness following kidney transplantation is as high as 40–80% [11,12].

A 2021 study by Raval et al. that analyzed data published over a 10-year period (2008–2018) presented the most up-to-date strategies used in the management of CMV infection, as well as the factors that contribute to the risk of CMV infection or disease. In addition, that study aimed to determine the impact caused by CMV infection. Descriptive statistical data from 69,803 adult kidney transplant recipients were analyzed and a pooled quantitative analysis was performed. That study found that prophylaxis and preventive therapy are the main strategies utilized for transplant recipients with CMV seropositive donor/seronegative recipient (D+/R−) and seropositive recipients (R+), respectively. Valganciclovir and ganciclovir are commonly used for patients with different risk levels for CMV infection. Despite implementing preventive measures, approximately 25% of renal transplant recipients develop CMV infection. Both age and the presence of a positive CMV D+/R− serostatus have been identified as consistent risk factors for CMV infection or disease. The presence of infection/disease was associated with higher mortality rates and a higher likelihood of transplant rejection. CMV has also been linked to an increased incidence of episodes of acute rejection, but there were significant variations between the included studies. That study also emphasized that CMV continues to be a substantial problem and that alternative approaches may be needed to increase the efficacy of the management of CMV infection in kidney transplant patients [13].

One of the most studied characteristics of CMV is the mechanism of viral reactivation, which leads to a range of clinically manifest syndromes, ranging from minor symptoms to tissue invasion, resulting in direct and indirect consequences in each patient [14]. Heald-Sargent’s study presents recent data indicating that ischemia/reperfusion was both necessary and sufficient to trigger CMV reactivation in a murine model after the transplantation of a graft that is latently infected. This study emphasizes that human and murine CMV reactivation is initially triggered by molecular events that activate CMV gene expression. Subsequently, lytic infection and virus spread are promoted by immunosuppression. The activation of viral gene expression can be triggered by oxidative stress, DNA damage, or inflammatory cytokines, and these factors may act together in a synergistic manner. The authors noted that novel therapeutic strategies are needed to effectively target this complex combination [15].

The Epstein–Barr virus (EBV) is a herpes virus that also serves as an important clinical entity: as a primary infection, it causes mononucleosis, but as an asymptomatic infection, EBV can affect various organs, including the respiratory, digestive, hematologic, and nervous systems, and can cause renal damage. In terms of nephropathic effects, symptoms include hematuria, proteinuria, nephrotic syndrome, and renal failure. EBV infections are poorly recognized, especially in children. Due to the variable severity and a lack of specific symptoms, EBV infections are often under-diagnosed and under-reported [16].

In kidney transplant patients, post-transplant lymphoproliferative disease (PTLD) is a very worrying complication that can occur after kidney transplantation. Over a 10-year period, people who have had a kidney transplant are 12 times more likely to develop PTLD than a similar group of people who have not had a transplant. Based on the frequency of kidney transplantation, kidney transplant patients who develop post-transplant lymphoproliferative disorder outnumber other organ transplant recipients who develop PTLD. A significant risk factor for PTLD is EBV infection, although 40% of PTLD cases in recent studies have not been related to EBV [17].

These three viruses are known to have deleterious effects in KTRs. Since 2011, the team of Koleilat I. et al. [18], from the USA, have screened for the viremia of three viruses (EBV/CMV/BKV) as an indicator of oncoming nephropathy, with subsequent reduction in immunotherapy, using PCR techniques. In their 134 transplanted patients, the authors detected BKV viremia in 16% cases, with no CMV or EBV involvement. The researchers concluded that the de-scalation of immunotherapy together with viremia evaluation every 30 days is safe and effectively prevents polyoma BK virus nephropathy. This therapeutic strategy was associated with a significantly decreased rate of CMV and EBV disease in KTRs, with no deleterious effects [18].

In a recent (2024) multicenter prospective observational study of KTRs from six USA transplant centers, Seifert M.E. et al. [19] analyzed 335 patients to define the natural history of BKV infection and identify the risk factors for BKV reactivation and BKV-associated nephropathy (BKVAN) in kidney transplant recipients. The authors found that persistent BK viremia/BKVAN was associated with poorer allograft function at 24 months after transplant. Their results may help design future clinical trials of therapies to prevent or mitigate the injurious impact of BKV reactivation on kidney transplant outcomes [19].

Zhao Y. et al. [20], from China, reported a simple, rapid, sensitive loop-mediated isothermal amplification (LAMP) assay using a high-fidelity DNA polymerase (HFman probe) for detecting BKV in urine, as it known that the early monitoring of BKV in urine is crucial to minimize the deleterious effects caused by this virus on the preservation of graft function. Their assay had high specificity and sensitivity (95% and 100%, respectively) and was combined with a portable finger-driven microfluidic chip for easy detection; this method shows great potential for the point-of-care testing of BKV [20].

In a pilot study, Gouvêa A.L. et al. [21], from Brazil, analyzed urinary decoy cells and PCR tests in samples from 32 consecutive kidney transplant patients to perform urinary screening for BKV reactivation. The authors found that early urinary monitoring is effective in the detection of BKV replication and represents a good strategy to minimize the effects caused by the presence of the virus on the preservation of graft function [21].

Similar studies have also been carried out in Europe.

Páez-Vega A. et al. [22], from Spain, conducted a randomized controlled trial, in which they evaluated whether it is effective and safe to discontinue prophylaxis when CMV-specific cell-mediated immunity (CMV-CMI) is detected and to continue with pre-emptive therapy. The authors found no difference in the incidence of CMV disease and replication in their 150 CMV-seropositive KT recipients, and they concluded that prophylaxis can be prematurely discontinued in CMV-seropositive KT patients receiving antithymocyte globulin when CMV-CMI is resolved, since no significant increase in the incidence of CMV replication or disease was observed [22].

In a retrospective multicentric trial, Boulay H et al. [23], from France, followed 372 CMV-seropositive renal transplant recipients for 3 years and found that CMV-associated disease occurred in 2.25% of patients in the prophylaxis (T-cell depleting induction group) and in 6% in the no-prophylaxis group. The incidence of allograft rejection and other infectious diseases was similar between the two groups. In their study, the authors found that the lack of prophylaxis had no deleterious effect on CMV-related diseases among CMV-seropositive renal transplant recipients receiving non-depleting induction [23].

In a review from 2015, Malvezzi P. et al. [24] stated that induction therapy with rabbit anti-thymocyte globulins prevents acute rejection after KT. However, because this induction therapy can have some harmful side effects (e.g., de novo post-transplant cancer or opportunistic CMV infections), the use of tacrolimus plus everolimus was suggested as a means to minimize these side effects and was found to be efficient in sensitized patients, in recipients from an expanded-criteria donor, and in patients where steroid avoidance is contemplated [24].

Mallat S. et al. [25] suggested ganciclovir and valganciclovir as IV antivirals to be used to manage the deleterious outcomes of CMV nephritis and to monitor CMV viremia with qPCR assays [25].

In a 5-year study that assessed the relationship between CMV infection and biopsy-proven graft rejection, Dmitrienko S. et al. concluded that, even though current antiviral therapy seems to mitigate the reported harmful effects of CMV infection on biopsy-proven acute rejection (BPAR) or graft survival, BPAR remains a significant risk factor for both CMV infection and functional graft survival [26].

The first paper identified in PubMed regarding the deleterious effects of CMV in KTR was published in 1985, by Bia M.J. et al. [27] from the Renal Transplant Service, Yale University School of Medicine. The authors concluded that therapies using antithymocyte globulin (ATG) to treat steroid-resistant rejections have a deleterious influence on the incidence and severity of CMV infection in renal transplant patients, even when the dosage of other immunosuppressive drugs is decreased during ATG therapy [27].

Aim: In this study, we performed a comprehensive evaluation of nephropathogenic viruses (BK, CMV, and EBV) in kidney transplant patients to assess the prevalence of infection with these viruses. In a future study, we will correlate the prevalence of nephritogenic viruses with different biomarkers, paraclinical parameters, as well as clinical data from patients in order to help establish a monitoring algorithm and personalized therapeutic approach.

## 2. Results

In our study, we detected 25/157 (16%) positive samples: 5 (3.2%) samples positive for EBV (FAM), 9 (5.7%) samples positive for CMV (VIC), and 11 (7%) samples positive for BKV (ROX). The Ct values and initial template quantity are shown in Table 1. By detecting three possible nephropathic and oncogenic viruses in KTRs simultaneously using multiplex real-time PCR, we aimed to optimize the monitoring and follow-up of these patients. This represents the originality of our study.

In Table 2, Table 3, Table 4 and Table 5 and Figure 1, Figure 2 and Figure 3, we present data about our tested patients in terms of gender distribution and each of the tested viruses in terms of viral load.

The EBV viral load values ranged from 234.4 × 10^2^ to 472.4 × 10^2^ DNA copies/mL.

The CMV-positive samples (9/157) showed varied viral loads: two patients with 10^1^ viral DNA copies, three with 10^2^ viral DNA copies, two with 10^3^ viral DNA copies, and one with the highest viral load of 10^4^ viral DNA copies.

The BKV-positive samples (11/157) showed varied viral loads: three samples with <10 viral DNA copies, six with 10^1^ viral DNA copies, one with 10^2^ viral DNA copies, and one with 10^3^ viral DNA copies.

## 3. Discussion

In our study, we tested 157 blood samples from kidney transplant patients for the presence of three viruses recognized for their role in transplant patient pathology: BKV, CMV, and EBV. The highest prevalence was detected for BKV (11/157, 7%), followed by CMV (9/157, 5.7%) and EBV (5/157, 3.2%).

The patients with positive samples received kidney grafts in the period from 2003 to 2024, with most transplants being carried out in the period from 2020 to 2023 and with two patients having received two consecutive transplants. The dates of CKD diagnosis fell between 1998 and 2022.

Four out of five EBV-positive patients received their kidneys from a cadaver donor. Two EBV-positive patients experienced previous rejection episodes, which were corticoid-sensitive, but one was diagnosed with chronic allograft disfunction. All five EBV-positive patients received immunosuppression therapy with tacrolimus, four received mycophenolate therapy, one sirolimus, and four prednisone. One patient had detectable EBV viremia, despite receiving valganciclovir therapy. Two patients experienced therapy-related side effects. One EBV-positive patient had a urothelial tumor prior to their transplant.

In the case of the CMV-positive patients, five kidney allografts were from cadaveric donors, while four were from living donors. Four patients had experienced a previous rejection episode, with one patient experiencing three consecutive rejection episodes. The rejection was corticoid-sensitive in three of these patients, and three were diagnosed with chronic allograft disfunction. Regarding immunosuppressive therapy in the CMV-positive patients, five were treated with tacrolimus, eight with mycophenolate, three with cyclosporine, and one with rituximab, due to previous medication side effects and biopsy results, which highlighted the membranous glomerulonephritis of the graft. Two patients with detectable CMV viremia underwent valganciclovir therapy. Two patients were previously diagnosed with CMV infection, with one receiving two consecutive transplants due to chronic graft rejection.

In the BKV-positive patient group, seven grafts were from a cadaver donor and two from a living donor. Two patients experienced previous rejection episodes, both being diagnosed with chronic allograft disfunction. Nine patients received tacrolimus and mycophenolate therapy, eight of whom also received prednisone and one of whom experienced digestive side effects and leukopenia. One patient received a transplant twice, with the first graft resulting in CMV-associated nephropathy, and another was previously diagnosed with a BKV infection. More available laboratory parameters are included in the Appendix A.

To compare our results with the data obtained by other authors, we analyzed articles published in PubMed in the last 5 years that tested the presence of BKV, CMV, and EBV using qPCR.

During this period, two papers were published on EBV qPCR testing in renal transplant patients. In 2020, Savassi-Ribas et al., from Brazil, detected EBV and CMV viremia in 32/82 renal transplant patients (representing 39% each) using TaqMan qPCR. The authors correlated EBV viremia with thymoglobulin administration, mTOR inhibitor therapy, and female gender and correlated CMV viremia with basiliximab administration, mycophenolate mofetil (MMF) therapy, and acute graft rejection at values above 10^4^ copies/mL. They found that EBV and CMV viremia is associated with different immunosuppressive therapy induction and maintenance strategies and that higher CMV loads (>10^4^ copies/mL) are related to acute graft rejection [28].

In 2019, Ashouri Saheli Z et al., from Iran, validated an in-house tetraplex nested PCR assay for the specific detection of BKV, JC polyomavirus (JCV), CMV, and EBV in clinical samples. The authors compared their assays with commercial uniplex qPCR kits and obtained good sensitivity values, with the highest values for CMV and EBV. The authors concluded that their nested PCR multiplex assay could be used as a reliable tool in post-renal-transplant surveillance [29].

We identified 12 studies [30,31,32,33,34,35,36,37,38,39,40,41] on the determination of CMV viremia through qPCR published in the last 5 years (Table 6).

We identified 13 studies [42,43,44,45,46,47,48,49,50,51,52,53,54] on the detection of BKV in kidney transplant patients using qPCR published in the last 5 years (Table 7).

In Table 6 and Table 7, we present 25 studies published in the last 5 years (2020–2024) that used real-time PCR assays to identify, follow up with, and guide therapy in CMV/BKV/EBV-positive KTRs.

The qPCR equipment and methods were very diverse, encompassing the cobas^®^ 6800 system; the StepOne™ device; the AmpliSense CMV-FL test system; TaqMan real-time PCR, establishing a level of 10^4^ CMV DNA copies/mL as the threshold for defining active infection; the CMV Real-RT Quant kit; the real-time PCR method using the CMV Real-TM Quant kit from Sacace, Como, Italy, and the Argene (bioMérieux, Marcy l’Etoile, France) system; the nanogen BK virus Q-PCR alert kit; the GeneProof BK/JC Virus PCR kit (GeneProof, Brno, Czech Republic); the RealStar BKV PCR Kit 1.0 (Altona Diagnostics GmbH, Hamburg, Germany); and the Applied Biosystems 7500 Real-Time PCR System (Applied Biosystems, Foster City, CA, USA). These methods ranged from in-house methods to fully automated systems. Although this shows the interest of companies in developing these detection and monitoring assays, useful not only in the case of transplant recipients but also for other medical pathologies (e.g., cervical cancer), these tests lack thorough clinical validation [55,56]. Each of these qPCR assays should be clinically validated in a significant number of cases to prove that they achieve optimal sensitivity, specificity, positive predictive value, and negative predictive value in comparison with the WHO guidelines, for example.

Furthermore, recent papers have been trying to use new modern methods to follow up with KTRs who are positive for BKV and CMV. Fernández-Ruiz M et al., from Spain, reported the utility of human microRNAs (hsa-miRNAs) as promising biomarkers to identify CMV-seropositive KTRs at risk of CMV reactivation despite detectable CMV-CMI [57]. Salinas T. et al. [58], from the USA, also reported urinary cell mRNA profiling for the non-invasive diagnosis of acute T cell-mediated rejection (TCMR) and BK virus nephropathy (BKVN), which could optimize patient management by minimizing the number of visits for urine collection [58]. In another study conducted in the USA, Sigdel TK et al. concluded that plasma proteomic and transcriptional perturbations impacting humoral and innate immune pathways are observed during CMV infection and provide biomarkers for CMV disease prediction and resolution [59]. All these new and modern methods should also be clinically validated in large cohorts and compared with official international data (e.g., from the WHO).

The 27 previously analyzed studies showed that there are numerous molecular methods of viral detection, applicable to different types of samples (whole blood, plasma, serum, urine, and biopsy samples). The cited studies had the following objectives: viral load detection for patient monitoring, the testing of tumor samples to find a link between polyomaviruses and oncogenesis, the analysis of viral genotypes in order to stratify patient risk, and the study of the role of different immune effectors in viral reactivation. We consider these studies to be proof of an increased interest in developing better monitoring protocols for kidney transplant recipients; however, future studies are needed to clarify the following questions: What is the most useful method of monitoring these patients? What are the major risk factors for kidney rejection? And, what would be the most effective preventive measures to decrease this risk while simultaneously increasing the quality of life of patients?

Regarding the most useful methods of monitoring KTRs, the current practice guidelines provide different recommendations for each of the three viruses regarding the frequency of testing, the possibility of prophylaxis, and long-term risks.

### Current Guidelines on the Evaluation of Kidney Transplant Patients

The Transplant Committee of the French Association of Urology (CTAFU) conducted a systematic review of the literature to study the prevalence, screening methods, diagnosis, and treatment of urothelial carcinoma in kidney transplant recipients and kidney transplant candidates. This study aimed to propose surgical guidelines for the treatment of urothelial carcinoma in individuals who received a kidney transplant or are potential candidates for transplantation [60].

The authors found that urothelial tumors are more common in people who have received a kidney transplant, with a three-fold higher rate of occurrence compared to the general population. Although the main risk factors for urothelial carcinomas are comparable to those in the general population, aristolocholic acid nephropathy and BK virus infection are more common in kidney transplant recipients. Kidney transplant recipients with non-muscle invasive bladder cancer have a higher and earlier rate of cancer recurrence compared to the general population. Retrospective data support the safety and efficacy of adjuvant intravesical treatments. The primary approach for the management of localized muscle-invasive bladder cancer in kidney transplant recipients is radical cystectomy. For renal transplant candidates with a history of urothelial cancer, it is essential to perform regular cystoscopies at the recommended frequency. The frequency of these follow-up procedures depends on the risk of tumor recurrence and progression. Regardless of the level of risk, the authors emphasize the need to continue these follow-up cystoscopies every six months until transplantation takes place. According to the most recent statistics, current guidelines provide specific parameters for the length of time that should elapse before active addition to the waiting list for renal transplant patients with a history of urothelial carcinoma [60].

The American Society of Transplantation—Infectious Diseases Community of Practice (AST-IDCOP) provides guidelines with up-to-date information on BK polyomavirus—associated infection, replication, and disease—particularly as it relates to kidney transplantation and, less often, to non-renal solid organ transplantation. Currently, there are no clinically validated measures for organ allocation, antiviral prophylaxis, or pre-transplant risk factor-based screening in kidney transplant donors and recipients. Therefore, it is recommended that all kidney transplant recipients be screened monthly for BK viremia up to nine months after transplant and then every three months to two years after transplant. Additional screening may be considered after 2 years for pediatric kidney transplantation. For renal transplant patients with plasma BKV DNA levels greater than 1000 copies/mL for 3 weeks or with levels increasing to over 10,000 copies/mL, reducing immunosuppression is indicated. These levels are associated with the likelihood of BKV-associated nephropathy [61]. This guideline states that the main strategy for treating biopsy-confirmed BKV-associated nephropathy is to reduce immunosuppression. It is therefore not necessary to perform allograft biopsy in the treatment of individuals with detectable BK viremia but who have normal renal function at baseline. Although there is scientific rationale to support it, there are no comprehensive randomized clinical trials that generally support switching from tacrolimus to cyclosporin-A or from mycophenolate to mTOR inhibitors or leflunomide; there is also no evidence for the use of intravenous immunoglobulin, leflunomide, or cidofovir as an add-on therapy. Prophylaxis or fluoroquinolone therapy is not suggested. Retransplantation may be successful in cases where the allograft loss is due to BKV nephropathy, as long as BK viremia is undetectable [61].

A study by the New England BK Consortium highlights the importance of BK polyomavirus, which continues to affect kidney transplant recipients. The main objective of this study was to increase the efficiency and effectiveness of BKV screening and management procedures. The authors aimed to analyze the BKV screening techniques of different centers and assess how they align with established consensus guidelines [62]. A survey was completed by thirteen of fifteen centers, representing 86.7% of the total. Only two centers adhered to the standards for BKV screening by utilizing monitoring parameters, while the remaining centers used less rigorous approaches and shorter durations. One center performed renal biopsies in all patients with plasma viral loads greater than 10,000 copies/mL, while the other centers performed biopsies only when there was a specific reason to do so. Eleven centers support performing a biopsy to confirm probable nephropathy. Twelve centers suggested a further reduction in immunosuppression to treat documented BKV-associated nephropathy. Nine medical centers indicated that their primary treatment involves dose reduction in calcineurin inhibitors. Over 50% of the evaluated centers reported using leflunomide, cidofovir, or intravenous immunoglobulin. The results show a significant variation in BKV screening and management approaches between different centers. Based on these findings, all centers involved agreed to adopt consistent screening procedures and to work towards improving the standards of management for this pathology [62].

A study by Imlay et al. analyzed BK virus nephropathy (BKVAN) as a significant problem that commonly affects kidney transplant recipients. This pathology usually develops in the first year after transplantation. The study noted that different guidelines provide different recommendations for screening for BKV after the first year and that no previous publication has provided a systematic review of the risk variables and outcomes associated with late-onset BKVAN (after >1 year) [63]. The team performed a retrospective analysis to compare the characteristics and outcomes of BKVAN occurring within the first year after transplantation (early onset) and BKVAN occurring after the first year (late onset) in a group of 671 patients who had undergone double kidney–pancreas transplantation at a single center in the United States between 2008 and 2013. BKVAN was confirmed either through biopsy or the detection of BK polyomavirus in plasma at a level greater than 10,000 copies/mL [63].

That study also found that BKVAN was identified in 96 (14.3%) people, with 16.7% having confirmed diagnoses and 83.3% having presumptive diagnoses. Of these cases, 79 (82.3%) were classified as early-onset and 17 (17.7%) as late-onset. The incidence of late-onset BKVAN was substantially higher in simultaneous kidney–pancreas transplant recipients compared with renal transplant patients (57.1% versus 14.6%). Of all cases, 82.4% exhibited de novo infection occurring after the first year without initial BKV detection, and 17.6% of cases exhibited the progression of a previous BKV reactivation. The clinical outcomes of BKVAN were comparable among the early- and late-onset patients. In a combined review of previous BKVAN research in simultaneous kidney–pancreas transplant recipients, 62.9% had late-onset symptoms. That study concluded that a sizable number of BKVAN cases occur later in life, particularly among simultaneous kidney–pancreas transplant recipients, and suggests that these patients should be monitored for BKVAN infection for a longer period than recommended in some guidelines [63].

It is important to standardize the identification and quantification of BKV to effectively manage transplant patients. This can be achieved by implementing global standards. Initially, the World Health Organization (WHO) international standard for the determination of BKV DNA (referred to as the WHO standard), was published in 2015 for this purpose. Subsequently, in 2018, a study by Engelmann et al. re-evaluated the WHO standard using a new automated BKV DNA detection assay called the kPCR PLX^TM^ BKV DNA assay (Siemens Healthcare Diagnostics, Marburg, Germany). In this research, the authors performed three sets of nucleic acid extraction and PCR assays in triplicate for different dilutions of the WHO standard (NIBSC code 14/212) [64] in both plasma and urine. The nucleic acid extraction process was performed using Versant kPCR SP molecular systems along with Versant 1.2 reagents for sample preparation. The PLX^TM^ BKV DNA kPCR assay was then performed on the Versant kPCR Molecular systems AD, manufactured by Siemens Healthcare Diagnostics in France. The team performed a linear regression analysis, which demonstrated a strong correlation between the expected and observed values for the WHO standard dilutions in both plasma and urine. A consistent and predictable bias was present in the WHO standard plasma dilutions. However, systematic bias was more visible for the WHO standard dilutions in urine compared to plasma. The factors for converting copies/mL to IU/mL were 1.67 for plasma samples and 0.33 for urine samples. The authors also cited a study that evaluated the Altona RealStar^TM^ BKV assay (Altona Diagnostics, Hamburg, Germany) and found the conversion factor for plasma samples to be 1.11 IU/copy. In contrast to the WHO standard dilutions in plasma, which were somewhat underestimated, the WHO standard dilutions in urine were found to be overestimated when using the PLX^TM^ BKV DNA kPCR assay, resulting in a conversion factor of 0.33, indicating the presence of a matrix effect. This event was previously documented in relation to the WHO guidelines for CMV and EBV [65].

An effective way to prevent cytomegalovirus infection during solid organ transplantation is not yet known. Pangonis et al. published a study reporting data on the incidence of CMV events after the implementation of a modified local prevention guideline that includes improved monitoring, earlier switching to oral valganciclovir, and the reduced use of immunoglobulin [66]. That study was based on a retrospective cohort analysis that was performed using historical controls to assess the incidence of CMV invasive disease before and after heart, liver, and kidney transplant surgery in pediatric patients. Data on the intervention outcomes were collected for the four-year period before and after the intervention, from September 2009 to October 2017. Logistic regression was used to calculate the probability of an incident associated with CMV infection. No detectable difference in the frequency of CMV invasive disease was identified between the two study cohorts. There was a significant increase in the detection of episodes of CMV infection, with the asymptomatic form being the most common. The observed increase was specifically linked to an increase in the frequency of surveillance testing among higher-risk heart and liver transplant patients, with an adjusted odds ratio of 1.08. Remarkably, 28.9% of CMV episodes occurred while patients were under the protection of antiviral treatment. The team found that modifying local CMV-associated disease prevention guidelines did not lead to an increase in invasive disease. Some episodes of CMV infection were observed during prophylaxis, indicating a possible distinction from adult solid organ transplant cases and highlighting the importance of monitoring the pediatric population during prophylaxis [66].

Current recommendations do not include a definition for monitoring CMV during the period of chronic infection. The long-term consequences of CMV infection, such as the presence of CMV in the blood without any symptoms, are not well understood. Ishikawa et al. performed a retrospective analysis at a single center to examine the occurrence of CMV infection during the chronic phase, which is defined as occurring more than 1 year after kidney transplantation. That study enrolled a total of 205 kidney transplant patients from April 2004 through December 2017 and tested CMV pp65 antigenemia at regular intervals of 1–3 months to detect the presence of CMV in the blood. In the chronic phase, 30.7% of individuals had asymptomatic CMV infections, while 2.9% developed CMV-associated disease. The results of this research indicate that the prevalence of CMV infection among patients after kidney transplantation remained constant at 10–20% per year over a 10-year period. CMV infection in the first year after kidney transplantation and chronic rejection were closely related to CMV viremia in the long term. There was a strong correlation between the presence of CMV in the blood during the chronic period and transplant failure. That study was the first to investigate the occurrence of CMV viremia 10 years after kidney transplantation. By taking measures to prevent latent CMV infection, the occurrence of graft rejection and graft loss following kidney transplantation could be reduced [67].

During the initial year following organ transplantation, recipients are most likely to develop PTLD. These diseases are mainly caused by infection with the Epstein–Barr virus. It is therefore essential to screen patients for anti-EBV antibodies before or at the time of transplantation. In uncommon cases (<5%) where the recipient does not have antibodies against EBV, there is a 95% chance that they will receive an organ from a donor who has antibodies against EBV. This increases the likelihood of developing a primary EBV infection and seroconversion shortly after transplantation. For such cases, it is recommended that the recipient undergo preventive antiviral therapy with acyclovir, valacyclovir, or ganciclovir. The Expert Group on Renal Transplantation (EBPG) provides a guideline stating that this medication should be started at the time of transplantation and continued for at least 3 months. The special suggestions provided for CMV prevention may be relevant in this scenario. The management of PTLD should be based on precise pathology, involving a thorough examination of cell markers and phenotyping. In all cases, basal immunosuppression should be reduced, either by maintaining steroid use only or by decreasing the doses of anti-calcineurin medication by at least 50% and discontinuing additional immunosuppressive drugs. For EBV-positive B-cell lymphoma, antiviral treatment with acyclovir, valacyclovir, or ganciclovir may be initiated for a minimum of one month or depending on the degree of EBV replication in the blood, given that this information is available. If a patient presents with rare lymphomas arising from mucosa-associated lymphoid tissue and tests positive for *Helicobacter pylori*, the complete clearance of the *H. pylori* infection using an approved methodology is required. The continued use of *H. pylori* prevention is recommended to prevent recurrence. For CD20-positive lymphomas, the recommended treatment is rituximab, a monoclonal antibody targeting CD20. Treatment involves giving one intravenous injection per week for a total of four weeks. For diffuse lymphomas or in cases where previous treatment has not been effective, CHOP chemotherapy should be given either alone or together with rituximab. The CHOP regimen consists of the drugs cyclophosphamide, doxorubicin, vincristine, and prednisone. It is important to consider stopping the use of immunosuppressive drugs completely, with or without the removal of the transplanted kidney [68].

At the Nephrology Department, Dialysis and Renal Transplant Center, “Dr. C.I. Parhon” University Hospital, patients are closely followed up with after transplant in order to assess the function of the allograft and to diagnose and try to prevent any complications related to immunosuppression in a timely manner. Patients are tested for viral infections, especially for those classically associated with high risk for transplant recipients (BKV and CMV), in accordance with this center’s protocols, which also include the control of the immunosuppressive medication level through periodic testing (by checking tacrolinemia, cyclosporinemia, etc.). This is in accordance with the previously cited guidelines; thus, the medication dose is carefully personalized to balance the risk of graft rejection with the risk of infectious complications, viral deleterious effects, and other medication-related side effects.

## 4. Materials and Methods

Our study included 157 renal transplant patients registered at the Clinical Hospital “Dr. C. I. Parhon”, Iași, Romania, between July 2024 and September 2024. All the renal transplant patients were included, excluding patients under 18 years of age, pregnant women, and patients unable to consent (Figure 4).

### 4.1. Triple Viral Detection by qPCR

Total DNA extraction was performed on 400 μL of blood, using the croBEE 201 a Nucleic Acid Extraction Kit (GeneProof, Brno, Czech Republic) and the automated croBEE NA16 Nucleic Acid Extraction System (GeneProof, Brno, Czech Republic), eluted in a final volume of 60 μL. The purified DNA was tested for the simultaneous detection of EBV, CMV, and BKV using the genesig^®^PLEX kit (PrimerDesign, Chandler’s Ford, UK). The assay utilized individual primers and probes designed for each virus, which were combined in a single reaction. Each virus was detected via different fluorescent channels (e.g., the FAM, VIC, and ROX channels for EBV, CMV, and BKV). The provided primer/probe mix exploited the TaqMan^®^ principle. During PCR amplification, the forward and reverse primers were hybridized to the target cDNA. The fluorogenic probes were included in the same reaction mixture, which consisted of a 5′ dye-labeled DNA probe and a 3′ quencher. During PCR amplification, the probe was cleaved, and the reporter dye and quencher were separated. The resulting fluorescence enhancement was detected using the Stratagene MX3005P qPCR platform (Agilent Technologies, Santa Clara, CA, USA). The assay utilized a positive control that provided an EBV signal through the FAM channel, a CMV signal through the VIC channel, and a BKV signal through the ROX channel. For the negative control, we added water without RNase/DNase. The endogenous control, which confirmed the extraction of a valid biological material, was the detection of the endogenous control through the Cy5 channel.

### 4.2. Preparing the Master Mix

For each DNA sample, we prepared a reaction mixture containing 10 µL of oasigPLEX master mix, 1 µL of Multiplex primer/probe mix, and 4 µL of RNase/DNase-free water. To this final volume of 15 µL, we added 5 µL of the tested DNA sample according to our experimental plate setup. The qPCR amplification protocol included an enzyme activation step, namely 2 min at 95 °C. We then used 50 cycles that included a denaturation step of 10 s at 95 °C and a data collection step of 60 s at 60 °C, according to the manufacturer’s instructions (PrimerDesign, Chandler’s Ford, UK) (Appendix A). Fluorogenic data were collected during this step through the FAM, VIC, ROX, and Cy5 channels. Our experiments were validated for the positive control (the FAM channel for EBV, the VIC channel for CMV, and the ROX channel for BKV) but not the template control (NTC) and endogenous control. The positive control signals indicated that this kit worked correctly when detecting each virus. The absence of any Ct in the NTC indicated that there was no cross-contamination during plate setup. Endogenous control: the signal obtained from the endogenous control reaction varied depending on the amount of biological material present in each sample. An early signal indicates the presence of a significant amount of biological material. A late signal indicates the presence of a small amount of biological material in the sample.

### 4.3. Preparation of the Standard Curve Dilution Series

Using our previous experience, we used Eppendorf tubes without RNase/DNase, added 90 μL of *template* solution *preparation buffer* to 5 of the tubes, and labeled them from 2 to 6. We added 10 μL of the positive control template into tube 2, then pipetted 10 μL from tube 2 into tube 3, and repeated steps 4 and 5 to complete the dilution series. Finally, we obtained the following results: tube 1—2 × 10^5^ per μL; tube 2—2 × 10^4^ per μL; tube 3—2 × 10^3^ per μL; tube 4—2 × 10^2^ per μL; tube 5—20 per μL; and tube 6—2 per μL of positive control DNA copies.

### 4.4. Data Interpretation

A sample was considered positive for EBV if a positive result was detected for the FAM target, for Cy5 (endogenous control), and for the positive control and a negative result was detected for NTC. A similar interpretation was made for CMV (VIC target) and BKV (ROX target). The positive control had to show amplification between Cq 16 and 23. This criterion is very important as failure to match the positive control could be a strong indication that the experiment was compromised (Figure 5 and Appendix A).

The parameters generated by the software (MX Pro version 3.0, build 311) of the qPCR instrument Mx 3005P (Agilent Technologies, Santa Clara, CA, USA) were within accepted limits:High efficiency: 97.5/94.9/101% (90–110%);Good accuracy: 0.995 (0.985–1.00);Y **=** −3.29 ((−3.1)–(−3.6)).

## 5. Conclusions

After testing 157 samples using triplex qPCR, we detected 25 samples positive for the presence of one of three viruses—CMV, EBV, or BKV—with a prevalence comparable to that found in other similar studies. Upon reviewing studies in the literature that have addressed this topic over the last 5 years, we confirmed that the use of real-time PCR for the detection of nephropathogenic viruses in kidney transplant patients is a robust, cost-effective method for case monitoring, especially in the first year after transplant. The differences in viral load are probably due to the prevention and personalized immunosuppression strategies applied in each center. These differences can be observed in the current guidelines, which provide several directions for the efficient monitoring of KTRs, but not all centers implement similar measures.

## Figures and Tables

**Figure 1 ijms-25-12698-f001:**
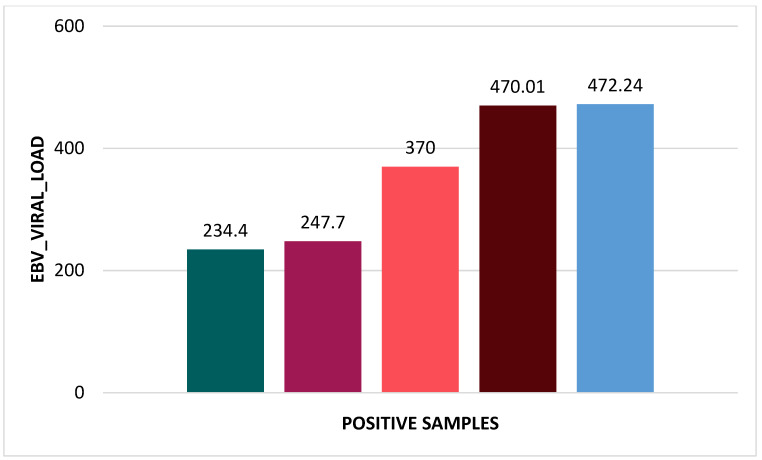
Viral load of the five patients positive for EBV.

**Figure 2 ijms-25-12698-f002:**
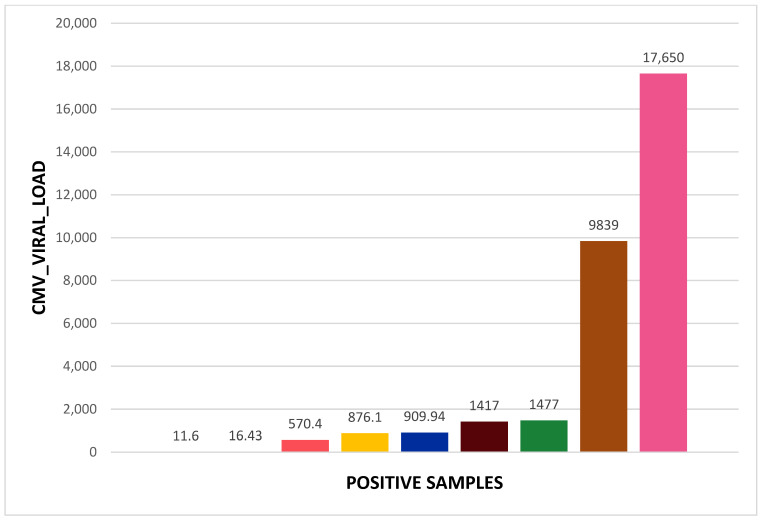
Viral load for the nine patients positive for CMV.

**Figure 3 ijms-25-12698-f003:**
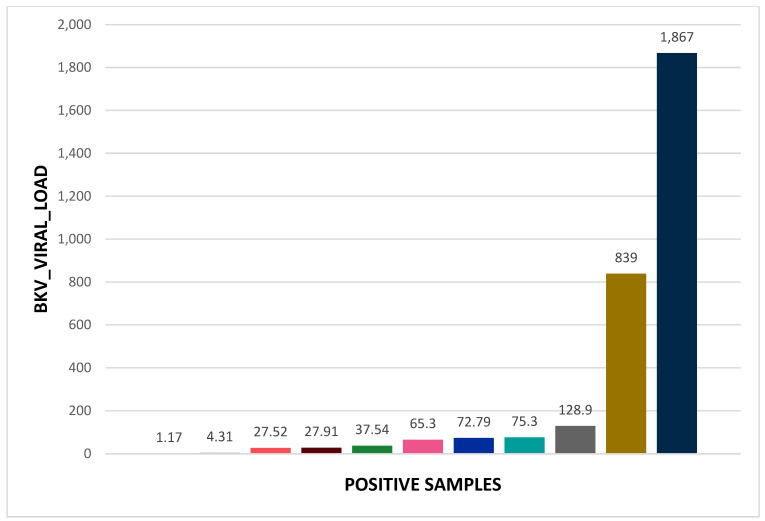
Viral load for the 11 patients positive for BKV.

**Figure 4 ijms-25-12698-f004:**
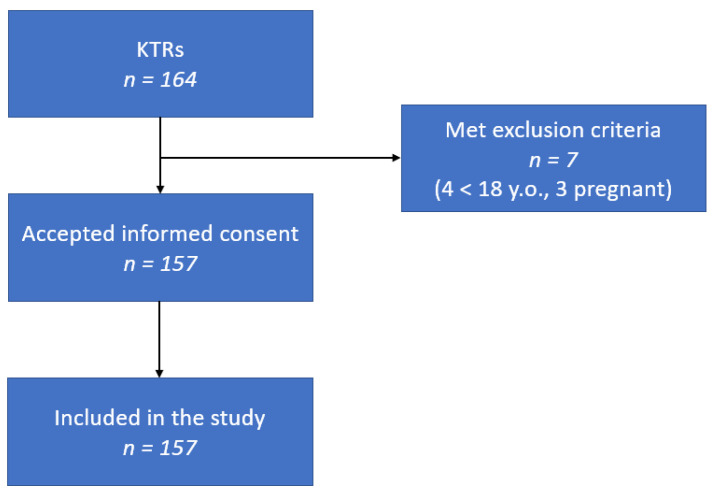
Diagram of the inclusion and exclusion criteria.

**Figure 5 ijms-25-12698-f005:**
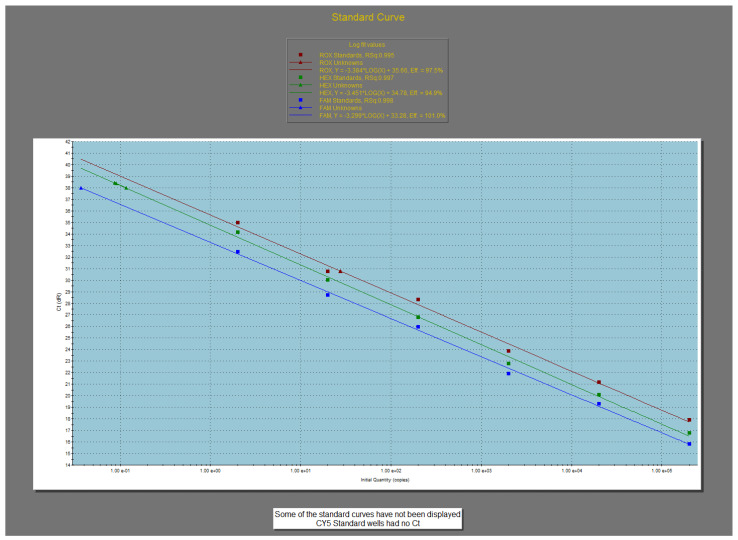
Standard curves for EBV/CMV/BKV.

**Table 1 ijms-25-12698-t001:** The Ct values and the initial template quantities of samples detected as being positive for oncogenic viruses.

FAM (EBV): 5 Positive Samples	Ct Value	Initial Template Quantity
1 (12)	38	3.7 × 10^2^
2 (52)	39	2.477 × 10^2^
3 (57)	38.08	4.724 × 10^2^
4 (67)	39	4.701 × 10^2^
5 (93)	39.2	2.344 × 10^2^
VIC Channel (CMV): 9 Positive Samples	Ct Value	Initial Template Quantity
1 (14)	38.43	8.761 × 10^2^
2 (24)	38	1.166 × 10
3 (26)	38.37	9.094 × 10^2^
4 (34)	38	5.704 × 10^2^
5 (68)	35.9	1.643 × 10
6 (77)	40	9.839 × 10^3^
7 (79)	38.75	1.417 × 10^3^
8 (85)	35.59	1.765 × 10^4^
9 (116)	38.73	1.477 × 10^3^
ROX Channel (BKV): 11 Positive Samples	Ct Value	Initial Template Quantity
1 (16)	30.77	2.791 × 10
2 (25)	30.79	2.752 × 10
3 (32)	36.38	7.530 × 10
4 (39)	35.82	1.17
5 (41)	33.86	4.31
6 (51)	32.87	8.39
7 (82)	40.46	7.279 × 10
8 (103)	42.57	1.289 × 10^2^
9 (104)	37.23	1.867 × 10^3^
10 (205)	36.1	6.530 × 10
11 (215)	31.59	3.754 × 10

**Table 2 ijms-25-12698-t002:** Distribution of patients by gender.

Gender					
		Frequency	Percent	Valid Percent	Cumulative Percent
Valid	Male	97	61.8	61.8	61.8
	Female	60	38.2	38.2	100.0
	Total	157	100.0	100.0	

Our sample included 157 patients, of whom 60 (38.2%) were women.

**Table 3 ijms-25-12698-t003:** Association between EBV and degree of viral load.

Crosstab							
Count							
		EBV_VIRAL_LOAD					Total
		234.40	247.70	370.00	470.01	472.24	
VIRAL_LOAD_CATEGORY	10^2^	1	1	1	1	1	5
Total		1	1	1	1	1	5

**Table 4 ijms-25-12698-t004:** Association between CMV and degree of viral load.

Crosstab										
Count										
		CMV_VIRAL_LOAD								Total
		11.66	16.43	570.40	876.10	909.94	1417.00	9839.00	17,650.00	
VIRAL_LOAD_CATEGORY	10^1^	1	1	0	0	0	0	0	0	2
	10^2^	0	0	1	1	1	0	0	0	3
	10^3^	0	0	0	0	0	1	1	0	2
	10^4^	0	0	0	0	0	0	0	1	1
Total		1	1	1	1	1	1	1	1	8

**Table 5 ijms-25-12698-t005:** Association between BKV and degree of viral load.

Crosstab													
Count													
		BKV_VIRAL_LOAD											Total
		1.17	4.31	27.52	27.91	37.54	65.30	72.79	75.30	128.90	839.00	1867.00	
VIRAL_LOAD_CATEGORY	<10	1	1	0	0	0	0	0	0	0	1	0	3
	10^1^	0	0	1	1	1	1	1	1	0	0	0	6
	10^2^	0	0	0	0	0	0	0	0	1	0	0	1
	10^3^	0	0	0	0	0	0	0	0	0	0	1	1
Total		1	1	1	1	1	1	1	1	1	1	1	11

**Table 6 ijms-25-12698-t006:** CMV detection using qPCR in KTRs.

No.	Author, Year, Country	Sample, Patients	Assay	Results	Clinical Importance
1	García-Campa M. et~al., 2024, México [30]	Blood samples from a 49-year-old patient who received a cadaver kidney transplant.	BK virus, B19 parvovirus, Epstein–Barr virus, and CMV were tested using real-time PCR.	The patient had undetectable viral loads for the four tested viruses. Subsequently, CMV infection was confirmed via gastrointestinal biopsy and immunohistochemistry.	This rare case, in which symptoms of such viral infection are present in the absence of a detectable viral load, draws attention to the importance of the clinical observation of patients. Histological analysis can be a diagnosis option for patients with undetectable viral loads.
2	Helary M. et~al., 2024, France [31]	A total of 390 paired plasma and whole blood samples from 60 hematopoietic stem cell transplant (HSCT) patients and 24 renal transplant patients.	CMV viremia levels were compared between the Cobas CMV plasma test (cobas^®^ 6800 system) and the reference Abbott RealTime CMV whole blood test (m2000 RealTime platform).	The sensitivity and specificity of plasma CMV load determination with the Cobas^®^ CMV assay were 90% in comparison to the whole blood Abbott assay.	The cobas^®^ CMV assay in plasma showed significant concordance with the Abbott RealTime CMV assay in whole blood, validating the significance of plasma samples for CMV surveillance in patients who underwent HSCT and kidney transplantation.
3	Zannat H. et~al., 2023, Bangladesh [32]	Serum samples from 32 patients within the first 6 months after transplantation.	The authors used a commercial kit for DNA extraction, and real-time PCR testing for CMV was performed on the StepOne™ device, also using a commercial kit.	In total, 11 samples were positive (34.4%) and 21 samples were negative (65.6%) for CMV.	The presence of CMV in one third of patients in the early post-transplant period draws attention to the importance of clinical and paraclinical monitoring during this critical period.
4	Vankova O.E. et~al., 2023, Russia [33]	Leukocyte mass, saliva, and urine samples from liver and kidney transplant patients.	The samples were tested for CMV DNA via real-time PCR using a commercial diagnostic kit (AmpliSense CMV-FL test system). CMV DNA samples were selected for genotyping, and the amplified DNA was used for sequencing and CMV genotyping, using the MiSeq sequencer (Illumina, San Diego, CA, USA). The *UL55*(gB) and *UL73*(gN) variable genes were used to determine the genotype.	Genotypes gB2, gN4c, and gN4b were dominant.	In some cases, two or three CMV genotypes from the same patient were identified, which shows the importance of the molecular diagnosis of this infection in kidney transplant patients and the usefulness of sequencing for obtaining rapid results with clinical impact.
5	Afshari A. et~al., 2022, Iran [34]	Blood samples from 61 transplant patients, 30 with active CMV infection and 31 with latent infection.	The patients’ viremia was measured using TaqMan real-time PCR, establishing a level of 10^4^ CMV DNA copies/mL as the threshold for defining active infection. To quantify the expression levels of certain target miRNA molecules, the authors utilized the SYBR Green real-time PCR principle.	There were significant increases in miR-UL112–3 p/5 p, -UL22A-3 p/5 p, -US25–1-5 p, -US25-2-3 p/5 p, -UL36-3 p-3 p/5 p, and -UL70-3 p in patients with active infection compared to those with latent infection.	The high expression levels of some miRNAs probably indicate the importance of these molecules in CMV pathogenesis and in the transition from latency to active infection.
6	Bodro M. et~al., 2022, Spain [35]	Whole blood samples from 116 pre-transplant seronegative (R-) cardiac, liver, pancreatic, or renal transplant patients who received grafts from seropositive (D+) donors.	In this study, the authors evaluated clinically and/or functionally relevant single-nucleotide polymorphisms (SNPs) (belonging to the TLR2, TLR3, TLR4, TLR7, TLR9, AIM2, MBL2, IL28, IFI16, MYD88, IRAK2, and IRAK4 regions). These were assessed using real-time PCR and sequence-based typing (PCR-SBT), and a score was derived to predict the occurrence of CMV infection and CMV-associated disease.	CMV infection occurred in 61 patients (53%) after a median duration of 163 days after transplantation; 33 patients had only asymptomatic viral replication (28%), 28 had CMV-caused disease (24%), and 11 (9%) had recurrent CMV infection. The score was able to predict CMV disease with a model AUC of 0.68, a sensitivity of 64.3%, and a specificity of 71.6%.	This study proved that an efficient model for predicting the risk of CMV disease in transplant patients (D+/R−) is feasible, but it requires several clinical validation steps.
7	Ahmed H.H. et~al., 2022, Sudan [36]	Blood samples from 104 kidney transplant patients.	Viral DNA was extracted using the QIAamp DNA mini kit and viral DNA amplification was performed using the CMV Real-RT Quant kit. Subsequently, the qB region was sequenced after nested PCR amplification.	CMV viremia was detectable in 40 patients (38.5%), with a mean value of 358 × 10^4^ copies/mL (values ranged from 62 copies/mL to 1.43 × 10^8^ copies/mL).	The association between type of immunosuppressive medication and high viral loads (>1000 copies/mL) was significant (*p* = 0.05), and the association between CMV viremia values >1000 copies/mL and CMV-associated disease symptoms was highly significant (*p* ≤ 0.001). The most common CMV genotype identified in the samples of these patients was gB3.
8	Soleimanian S. et~al., 2022, Iran [37]	Blood samples from 10 KTRs with CMV DNA levels > 10^4^ copies/mL.	The CMV DNA levels in blood were determined using real-time PCR. NK cell phenotype was determined using flow cytometry.	Increased CMV viremia was negatively correlated with the presence of CCR7 + CD57+ CD56/CD16+ NK lymphocyte subtypes, whereas CCR7-CD57- CD56/CD16+ subtypes were positively correlated with CMV viremia.	The expression of CCR7 is linked to CMV reactivation, presenting a novel aspect of CMV-related immunity within the NK cell population.
9	Taksinwarajarn T. et~al., 2021, Thailand [38]	Plasma samples from 494 renal transplant patients.	CMV viremia, determined using real-time PCR, was detectable in all plasma samples from the patients, with a mean value of 11,102 IU/mL; the authors obtained a cut-off value of 4063 IU/mL for the diagnosis of CMV-associated disease.	Only 17 (3.4%) patients had gastrointestinal manifestations of CMV-associated disease. Among them, 15 had D+/R+ status and 14 were within the first 6 months after transplant.	The application of very sensitive nucleic acid amplification tests may facilitate the detection of CMV gastrointestinal illness in individuals with a CMV D+/R+ serostatus. KTRs with CMV seromismatch and extended cold ischemia time face an elevated risk of CMV gastrointestinal disease.
10	Taher N.M. et~al., 2021, Iraq [39]	Serum samples from 80 kidney transplant recipients, more than 6 months after the transplant.	DNA extraction from serum samples was performed using the QIAamp^®^ DNA Mini Kit (Qiagen, Hilden, Germany). The quantification of CMV viremia was performed via a real-time PCR method using the CMV Real-TM Quant kit from Sacace, Como, Italy. The qualitative detection of TTV was carried out using the Bosphore^®^ TTV detection Kit v1.	CMV viremia was detectable in 25% of samples, and TTV viremia in 56.25% of samples. The TTV antigen was determined to be positive based on ELISA in 10% of transplant patients. CMV was detected in only 20% of TTV-positive samples.	TTV is not associated with CMV reactivation in KTRs, and the presence of the TTV antigen has a significant association with low IL-6 levels and thus indicates a lower risk of rejection in these patients.
11	Minz R.W. et~al., 2020, India [40]	A total of 1610 blood samples from patients with suspected CMV disease out of a total of 2681 renal transplants.	The assays used were pp65Ag testing and CMV DNA detection via real-time PCR.	The incidence of CMV-associated syndrome was 14.25%, while viremia was detectable in 23.73% of patients with clinical suspicion.	These data show that CMV infection is a highly prevalent infection among renal transplant patients and, therefore, rapid diagnosis, together with immediate treatment or the use of prophylaxis, has a very important role in the management of post-transplant cases. The authors also argue that quantitative methods such as real-time PCR are the gold standard in this diagnosis.
12	Pérez-Flores I. et~al., 2019, Spain [41]	Blood samples from 498 renal transplant patients.	CMV was detected via real-time PCR using the Argene (bioMérieux, Marcy l’Etoile, France) system. DNA extraction was performed using the MagNA Pure automated system, followed by genotyping of the *IL-18* promoter region.	CMV infection was diagnosed in 38% of patients, and CMV-associated disease in 7.5%. Patients presenting the -607 C/-137 G haplotype and receiving antiviral prophylaxis had an increased incidence of CMV replication when prophylaxis was discontinued.	This haplotype was associated with a higher rate of CMV replication, after prophylaxis.

**Table 7 ijms-25-12698-t007:** BKV detection by qPCR in KTRs.

No.	Author, Year, Country	Sample, Patients	Assay	Results	Clinical Importance
1	Sahragard I. et~al., 2023, Iran [42]	Blood samples from 47 KTRs. The selected group of participants was divided into patients with active and inactive BKV infection.	For CMV detection, the team used the BKPyV TaqMan real-time kit (GeneProof, Brno, Czech Republic) on the Step One Plus (Applied Biosystems, USA) platform. The expression level of some transcription factor genes was determined using the SYBR Green real-time PCR principle.	The expression level of the transcription factor genes *SP1, NF1, SMAD, NFκB, P53, PEA3, ETS1, AP2, NFAT,* and *AP1* were significantly increased in patients with active infection compared to those with inactive infection.	The results indicated that there was a significant correlation between viral load level and mutation frequency.
2	Hamed R. et~al., 2023, Saudi Arabia [43]	Plasma samples in a group of 81 pediatric kidney transplant patients.	Periodic BK viremia determination was performed using the nanogen BK virus Q-PCR alert kit.	Patients diagnosed with BK nephropathy had a higher viral load than BK viremia patients.	BK viremia and BK nephropathy showed similar occurrence rates between children and adults.
3	Iwasaki S. et~al., 2023, Japan [44]	Tumor samples from a patient with urothelial carcinoma of the bladder, 54 months after kidney transplantation.	To highlight the oncogenic role of BKV, the authors detected BKV DNA and mRNA encoding the BKV LT antigen in tumor tissue samples using real-time PCR.	The target was a sequence of the hypervariable region of the BKV genome–VP1, which was detected both in the renal graft biopsy (performed because the patient had BKV nephropathy) and in the urothelial carcinoma tissue, suggesting that the BKV-associated tumor developed under persistent nephropathy.	Because of the rapidly progressive nature of urothelial carcinomas, close monitoring through the evaluation of urinary cytology and cystoscopy is required.
4	Li C. et~al., 2022, USA [45]	A total of 192 paired urine samples and 192 renal allograft biopsies from 155 patients.	The BKV VP1 mRNA copy number was identified using qRT-PCR.	The novelty of the study was the use of a murine amplicon of the *Bak* gene for the calibration curve of the reaction, with highly favorable results (a slope of −3.291, a Y-intercept of 38.60, an R2 value of 1.00, an efficiency of 101%, and an error of 0.014).	The authors believe that this Bak standard curve could serve as a universal calibrator for the absolute quantification of transcripts in qRT-PCR assays and would have the advantage of reducing labor and cost and eliminating the contamination of genes of interest through the repeated amplification of gene-specific standard curves.
5	Keykhosravi S. et~al., 2022, Iran [46]	Plasma and urine samples from 120 kidney transplant patients.	The simultaneous detection and quantification of JCV DNA and BKV DNA in plasma and urine samples using real-time PCR was performed using GeneProof™ real-time PCR kits (GeneProof, Brno, Czech Republic) on the StepOne Plus™ instrument (Applied Biosystems, Foster City, CA, USA).	The mean load of the urinary JCV was ten times more than that of the plasma viral load. There was a significant association between BKV and JCV viremia among tested patients. The frequency of JCV in men was almost three times higher than that in women.	The renal replication of BKV favors the urinary shedding of JCV. The excretion of JCV in urine was significantly associated with steroids such as prednisolone acetate.
6	Komorniczak M. et~al., 2022, Poland [47]	Urine samples from 155 patients who received cadaver kidney transplants.	The samples were tested via real-time PCR for the detection and quantification of BKV and JCV using the GeneProof BK/JC Virus PCR kit (GeneProof, Brno, Czech Republic) and using LightCycler 480 (Roche, Basel, Switzerland). The replication of the two polyomaviruses was recognized when the virulence level was >10^7^ viral DNA copies/mL of urine.	BKV/JCV infection was diagnosed in 68 cases (43.59%), with 31 patients (20%) positive for BKV, 35 patients (22.6%) positive for JCV, and 2 cases (1.25%) of coinfection. Based on the results obtained, patients were divided into three groups: group 1 had 87 patients (56.1%) with undetectable viruria; group 2 had 44 patients (28.4%) with detectable viruria but without polyomavirus replication; and group 3 had 24 patients (15.5%) with active viral replication.	The presence of viruria was correlated with the type of immunosuppressive regimen, strongly associated with tacrolimus administration. BKV/JCV viruria may be a good screening marker for active polyomavirus infection, as well as a predictor of nephropathy.
7	Rahimi Z. et~al., 2022, Iran [48]	Plasma samples from 31 renal transplant patients with active BKV infection: 32 patients with inactive BKV infection and 30 healthy individuals as the control group.	Viral BK load was tested using the BKV TaqMan real-time PCR kit (PrimerDesign, Chandler’s Ford, UK). Gene expression levels for the cytokines IL-27, IFN-γ, TNF-α, TNFR2, and IRF7 were also assessed via SYBR Green PCR.	The expression level of these cytokines was significantly higher in the group with inactive BKV infection compared to the group with active infection.	The BKV-active group exhibited considerably lower levels of IL-27 and IRF7 compared to the inactive group. This suggests that further research in this area is necessary to fully understand the regulatory role of BK virus infection in kidney transplant recipients.
8	Sommerer C. et~al., 2021, Germany [49]	Whole blood samples of 64 patients from three centers in Europe.	Frequent monitoring of the patients was carried out to detect BKV, CMV, and EBV. The researchers evaluated the expression of nuclear factor-regulated T-lymphocyte-activating factor-regulated genes (NFAT-RGE) in patients who received renal allografts as a predictive biomarker for identifying patients at risk of acute rejection or infection. NFAT-RGE (IL-2, IFN-γ, GM-CSF) were assessed via real-time PCR.	Patients with elevated residual gene expression (NFAT-RGE ≥ 30%) had an increased risk of acute graft rejection during the months following transplant. Patients with low residual gene expression (NFAT-RGE < 30%) had an increased incidence of infectious complications, especially CMV and BKV replication.	In the early post-transplant phase, NFAT-RGE was demonstrated to be a viable early predictive biomarker for identifying patients at risk of acute rejection and infectious side effects from tacrolimus medication.
9	Muñoz-Gallego I. et~al., 2021, Spain [50]	A total of 180 blood samples with detectable BK viremia > 1000 copies/mL from 63 kidney transplant recipients.	The detection of BKV viremia was performed using the RealStar BKV PCR Kit 1.0 (Altona Diagnostics GmbH, Hamburg, Germany), and BKV genotypes were determined using rt-PCR in a LightCycler 2.0 thermal cycler (Roche, Basel, Switzerland).	The most common genotypes were I, in 14 patients (22.2%), and II, in 13 patients (20.6%). Half of the patients in the study had multiple genotypes, with the majority of them having genotypes I and II.	These data on BKV genotypes may be relevant in stratifying the risk of BKV-associated complications and guiding the clinical management of BKV infection in kidney transplant patients.
10	Signorini L. et~al., 2020, Italy [51]	Urine samples from 57 kidney transplant donor/recipient pairs.	Urine was tested for the presence of BKV, JCV, and Merkel cell polyomavirus (MCV) DNA using qPCR on the Applied Biosystems 7500 Real-Time PCR System (Applied Biosystems, Foster City, CA, USA) with specific primers and TaqMan probes targeting the VP1 sequence. Subsequent molecular characterization was carried out via sequencing.	Genomes of the three polyomaviruses were detected in 49.1% of donors and 77.2% of recipients. The sequencing data revealed the archetypal strain for JCV, the TU and Dunlop strains for BKV, and the IIa-2 strain for MCV. VP1-based genotyping showed increased frequency of the JCV genotype 1 and BKV genotype I.	The authors emphasize that the genomes of polyomaviruses remain stable over time after transplantation without the occurrence of quasispecies. The strains isolated in the donor/recipient pairs were mostly identical, supporting that the viruses detected in the recipient may be graft-transmitted.
11	Bertz S. et~al., 2020, Germany [52]	Five urothelial tumor samples.	This research team tested the presence of BKV DNA using PCR in urothelial tumor samples and the occurrence of variations in the tumor and viral genomes using NGS. Immunohistochemistry was used to screen for the presence of LT antigens in 94 micropapillary carcinoma (MPUC) samples, 480 undifferentiated urothelial carcinoma samples, 199 muscle invasive carcinoma samples (including 83 with variant differentiation), 76 cases of plasmacytoid tumors, and 15 post-transplant urothelial carcinomas.	Five tumors met the criteria for inclusion in later stages of the study. BKV DNA was detected in 5/5 urothelial carcinomas, and the chromosomal integration of the BKV genome was identified in 4/5 such tumors.	The authors argue that LT antigen expression by integrated BKV genomes leads to p53 inactivation and aggressive, high-grade tumors with unusual, often micropapillary, morphology.
12	de Almeida S.G.S. et~al., 2020, Brazil [53]	Preliminary study: 130 plasma samples from 33 kidney transplant recipients in a preliminary study. Implementation study: 472 plasma samples from 84 transplant patients.	These samples were analyzed for BKV viremia using a qualitative semi-nested PCR (snPCR) method, followed by the quantification of BKV viremia after transplantation (qPCR).	Preliminary results: eight plasma samples from six patients showed detectable BKV via qPCR (18%), and seven of these samples were also detected as being positive through the snPCR assay (sensitivity = 88%). Implementation study results: 28 samples (6%) were detected as being positive using snPCR and the qPCR assay confirmed BKV DNAemia in 26 of them (5.5%). A total of 25% of patients had at least one sample with BKV- DNAemia confirmed via qPCR. A total of 20% of patients exhibited detectable BKV in only one sample, while 5% had two or more positive samples.	In patients with detectable BKV viremia, no kidney graft loss occurred. Cost assessments indicated that in resource-limited scenarios, the snPCR technique would be a more affordable option for BKV DNAemia screening.
13	Querido S. et~al., 2020, Portugal [54]	Biopsy samples from high-grade urothelial carcinoma, developed 9 years after kidney transplantation. Periodic urine samples for follow-up after transplantation.	JCV and BKV viremia and viruria were tested periodically during post-transplant monitoring via qPCR, using the LightMix Kit for the differential detection of JCV and BKV Polyomaviruses (Roche, Basel, Switzerland). The biopsy samples were tested for BKV and JCV DNA using qPCR. Immunohistochemistry was used to highlight the presence of the polyomavirus LTAg.	Biopsy samples from the neoplastic tissue were positive for JCV DNA and negative for BKV DNA. Immunohistochemistry showed strong positivity for the cell cycle markers p16, p53, and Ki67 and for the early JCV viral protein, LT antigen (present in polyomavirus species), while one of the late viral proteins—VP1—was stained negative, indicating a cell cycle blockage between the two events with accumulation of the LT antigen.	This study presented the first case of high-grade urothelial carcinoma associated with JCV nephropathy in a renal transplant patient.

## Data Availability

The original contributions presented in the study are included in the article/Appendix A, further inquiries can be directed to the corresponding author.

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
