# Peer review of "Accurate Multiplex qPCR Detection of Epstein–Barr Virus/Cytomegalovirus/BK Virus in Kidney Transplant Patients: Pilot Study"

_ijms, 2024, doi:10.3390/ijms252312698_

Round 1

Reviewer 1 Report

Comments and Suggestions for Authors

Dear authors, thank you very much for allowing me to read your work.

You are describing the utilization of RT-PCR for the detection of BKV, CMV, and EBV infection in renal transplant recipients, in a relatively large transplant cohort.

I have the following comments:

* The manuscript body should be focused in the interrelation between viral load (EBV, CMV, BKV), and relevant clinical outcomes with the use of RT-PCR. Focus should be given concerning therapeutic intervention guidance through RT-PCR surveillance.

*Please describe testing inclusion and exclusion criteria. Provide a diagrammatic representation.

* Please provide the patient-relevant information regarding the time since transplantation, immunosuppressive treatment, rejection episodes, times transplanted, presence of cytotoxic antibodies, renal function parameters (serum creatinine, proteinuria), and renal biopsy parameters. Are the above parameters related to RT-PCR results?

*Minor comment: Figure 4,5,6,7 should be provided as supplementary material.

All the best.

Author Response

Point-by-point response to the reviewers’ and editor’s comments

Title: “Accurate multiplex qPCR detection of EBV / CMV / BKV in kidney transplant patients: pilot study”

We thank the reviewer for giving us the opportunity to improve the quality of our manuscript.

Reviewer #1:

Comments and Suggestions for Authors

Dear authors, thank you very much for allowing me to read your work.

You are describing the utilization of RT-PCR for the detection of BKV, CMV, and EBV infection in renal transplant recipients, in a relatively large transplant cohort.

I have the following comments:

* The manuscript body should be focused on the interrelation between viral load (EBV, CMV, BKV), and relevant clinical outcomes with the use of RT-PCR. Focus should be given concerning therapeutic intervention guidance through RT-PCR surveillance.

o    Response to Reviewer: We thank the reviewer for this comment. We have cited more studies and discussed the interrelation between the viruses, relevant clinical outcomes and real-time PCR monitoring, as suggested (lines 347-376). Moreover, we provided some data regarding patient monitoring in Nephrology Department, Dialysis and Renal Transplant Center, “Dr. C.I. Parhon” Uni-versity Hospital (lines 575-585):

In Tables 6 and 7 we presented 25 studies which have been published in the last 5 years (2020 – 2024), which used Real Time PCR assays to identify, follow up and guide therapy in CMV / BKV / EBV positive KTRs.

The qPCR equipments and methods were very diverse - cobas® 6800 system, StepOne™ device, AmpliSense CMV-FL test system, Taq-man real-time PCR, establishing a level of 104 CMV DNA copies/mL as the threshold for defining active infection, CMV Re-al-RT Quant kit, real-time PCR method using the CMV Real-TM Quant kit from Sacace, It-aly, using the Argene (bioMérieux) system, nanogen BK virus Q-PCR alert kit, GeneProof BK/JC Virus PCR kit (GeneProof a.s., Czechia), RealStar BKV PCR Kit 1.0 (Altona Diagnos-tics GmbH, Germany), Applied Biosystems 7500 Real-Time PCR System (Applied Biosys-tems, USA), from in house methods, to fully automated systems. This shows the interest of companies for developing these detection and monitoring assays, useful in the case of transplant recipients, but similarly to other medical pathologies (e.g., cervical cancer), these tests lack thorough clinical validation [55, 56]. Each of these qPCR assay should be clinically validated on a significant number of cases and to prove that they get optimal sensitivity, specificity, positive predictive value and negative predictive value, in compar-ison with WHO guideline, for example.

More than this, recent papers are trying to use new modern methods to follow up KTRs who are positive for BKV and CMV. Fernández-Ruiz M et al., from Spain are report-ing the utility of human microRNAs (hsa-miRNAs) as promising biomarkers to identify CMV-seropositive KTRs at risk of CMV reactivation despite detectable CMV-CMI [57]. Sa-linas T et al., from USA, have also reported a urinary cell mRNA profiling for non-invasive diagnosis of acute T cell mediated rejection (TCMR) and BK virus nephropathy (BKVN), which could optimise patient management by minimizing their visits for urine collection [58]. In another study conducted in the USA, Sigdel TK et al. concluded that plasma pro-teomic and transcriptional perturbations impacting humoral and innate immune path-ways are observed during CMV infection and provide biomarkers for CMV disease predic-tion and resolution [59]. All these new and modern methods should also be clinically validated, on large cohorts, in comparison with official international data (e.g., WHO).

In the practice of Nephrology Department, Dialysis and Renal Transplant Center, “Dr. C.I. Parhon” University Hospital, patients are closely followed up after transplant, regarding the function of the allograft, but also to timely diagnose and try to prevent any kind of complications related to immunosuppression. Patients are tested for viral infec-tions, especially for those classically associated with high risk for transplant recipients (BKV and CMV), according to this center’s protocols, that include also the control of the immunosuppressive medication level that is periodically tested (by checking tacro-linemia, cyclosporinemia etc.). This is in accordance to the previously cited guidelines, in this manner, the medication dose is carefully personalized, to balance the risk of graft re-jection with the risk of infectious complications and viral deleterious effects, and other medication-related side effects.

*Please describe testing inclusion and exclusion criteria. Provide a diagrammatic representation.

o    Response to Reviewer: We thank the reviewer for this comment. We provided a diagrammatic representation in Figure 4, as suggested (lines 590-592):

* Please provide the patient-relevant information regarding the time since transplantation, immunosuppressive treatment, rejection episodes, times transplanted, presence of cytotoxic antibodies, renal function parameters (serum creatinine, proteinuria), and renal biopsy parameters. Are the above parameters related to RT-PCR results?

o    Response to Reviewer: We thank the reviewer for this comment. Although in this pilot study we focused on the monitoring method, we provide the requested patient information, for the positive patients, as suggested (lines 292-321 + Supplementary material 2):

The patients with positive samples have received the kidney graft in the period 2003-2024, with most transplants being done in the period 2020-2023, and two patients had received two consecutive transplants. The date of CKD diagnosis ranged from 1998 to 2022. 

4/5 EBV-positive patients received their kidney from a cadaver donor. Two EBV-positive patients experienced previous rejection episodes, which were corti-coid-sensitive, but one was diagnosed with chronic allograft disfunction. All 5 EBV-positive patients received immunosuppression therapy with tacrolimus, 4 received mycophenolate therapy, 1 sirolimus and 4 prednisone. One patient had detectable EBV viremia, despite receiving valganciclovir therapy. Two patients experienced thera-py-related side effects. One EBV-positive patient had a urothelial tumor prior to being transplanted.

In the case of the CMV-positive patients, 5 kidney allografts were from cadaveric donors, while 4 were from living donors. 4 patients had experienced a previous rejection episode, with one patient experiencing 3 consecutive rejection episodes. The rejection was corticoid sensitive in 3 of these patients, and 3 were diagnosed with chronic allograft dis-function. Regarding immunosuppressive therapy of the CMV-positive patients, 5 were treated with tacrolimus, 8 with mycophenolate, 3 with cyclosporine and one with rituxi-mab, due to previous medication side effects and biopsy results, which highlighted mem-branous glomerulonephritis of the graft. Two patients with detectable CMV viremia were under valganciclovir therapy. 2 patients were previously diagnosed with CMV infection, and one from this group received 2 consecutive transplants, due to chronic graft rejection.

From the BKV-positive patients group, 7 grafts were from a cadaver donor and 2 from a living donor. 2 patients experienced previous rejection episodes; both being diagnosed with chronic allograft disfunction. 9 patients received tacrolimus and mycophenolate therapy, 8 of which associated prednisone, and one patient experienced digestive side ef-fects and leukopenia. 1 patient was transplanted twice, with the previous graft experienc-ing CMV-associated nephropathy and another was previously diagnosed with a BKV in-fection. More available laboratory parameters were included in Supplementary material 2.

*Minor comment: Figure 4,5,6,7 should be provided as supplementary material.

o    Response to Reviewer: We thank the reviewer for this comment. We prepared the supplementary material, with figures 4 – 7, as suggested (line 620).

Reviewer 2 Report

Comments and Suggestions for Authors

This interesting study analyzes viremia of 3 pathogenic viruses in renal transplant patents namely CMV, EBV and BK virus by a triplex PCR method.

I suggest to the authors to have a more concise introduction focusing on the deleterious effect of these 3 viruses in renal transplant as their respective epidemiology and way of detection. Similarly, their discussion should focus on their results , comparison with data of literature with similar or close methodology , the potential  interest for monitoring patients as the gain of expense.

Comments on the Quality of English Language

It would be wise to check the English by a UK ou US or commonwealth native translator

Author Response

Point-by-point response to the reviewers’ and editor’s comments    

Title: “Accurate multiplex qPCR detection of EBV / CMV / BKV in kidney transplant patients: pilot study”  

We thank the reviewer for giving us the opportunity to improve the quality of our manuscript.  

Reviewer #2:   Comments and Suggestions for Authors  

This interesting study analyzes viremia of 3 pathogenic viruses in renal transplant patents namely CMV, EBV and BK virus by a triplex PCR method.  

1. I suggest to the authors to have a more concise introduction focusing on the deleterious effect of these 3 viruses in renal transplant as their respective epidemiology and way of detection.  

Response to Reviewer: We thank the reviewer for this comment. We have tried to described in detail the deleterious effects of the three viruses, in the introduction, as suggested (lines 172-236):

These three viruses are known to have deleterious effects in KTRs. Since 2011, the team of Koleilat I et al., from the USA, conducted a screening for viremia of 3 viruses (EBV / CMV / BKV viruses) as an indicator of oncoming nephropathy, with subsequent reduction in immunotherapy, using PCR techniques. In their 134 transplanted patients, the authors detected BKV viremia in 16% cases, with no CMV or EBV involvement. The researchers concluded that de-scalation of immunotherapy together with viremia evaluation every 30 days, is safe and effectively prevents Polyoma BK virus nephropathy. This therapeutical strategy was associated with a significantly decreased rate of CMV and EBV disease in KTRs, with no deleterious effects [18].

Seifert ME et al, in a recent (2024) multicenter prospective observational study of KTRs, from six USA transplant centers, analysed 335 patients to define the natural history of BKV infection, identify risk factors for BKV reactivation and BKV-associated nephropathy (BKVAN) in kidney transplant recipients. The authors found that persistent BK viremia/BKVAN was associated with poorer allograft function by 24 months post-transplant. Their results may help design future clinical trials of therapies to prevent or mitigate the injurious impact of BKV reactivation on kidney transplant outcomes [19].

Zhao Y, et al, from China had reported a simple, rapid, sensitive loop-mediated isothermal amplification (LAMP) assay using a high-fidelity DNA polymerase (HFman probe) for detecting BKV in urine, as it known that early monitoring of BKV in urine is crucial to minimize the deleterious effects caused by this virus on preservation of graft function. Their assay had high specificity and sensitivity (95% and 100% respectively) and combined with a portable finger-driven microfluidic chip for easy detection, this method shows great potential for point-of-care testing of BKV [20].

In a pilot study, Gouvêa AL et al., from Brasil, analysed urinary decoy cells and PCR tests in samples from 32 consecutive kidney transplant patients, to perform urinary screening for BKV reactivation. The authors found that early urinary monitoring is effective in detection of BKV replication and it represents a good strategy to minimize the effects caused by the presence of the virus on preservation of graft function [21].

Similar studies have also been carried out in Europe:

Páez-Vega A et al., from Spain, conducted a randomized controlled trial, in which they evaluated whether it is effective and safe to discontinue prophylaxis when CMV-specific cell-mediated immunity (CMV-CMI) is detected and to continue with pre-emptive therapy. The authors found no difference in the incidence of CMV disease and replication, in their 150 CMV-seropositive KT recipients, and they concluded that prophylaxis can be prematurely discontinued in CMV-seropositive KT patients receiving antithymocyte globulin, when CMV-CMI is resolved, since no significant increase in the incidence of CMV replication or disease is observed [22].

In a retrospective multicentric trial, Boulay H et al., from France, followed 372 CMV-seropositive renal transplant recipients for 3 years, and they found that CMV-associated disease occurred in 2.25% of patients in the prophylaxis (T-cell depleting induction group) and in 6% in the no-prophylaxis group. The incidence of allograft rejection and other infectious diseases was similar between the two groups. In their study, the authors found that the lack of prophylaxis had no deleterious effect for CMV-related diseases among CMV seropositive renal transplant recipients receiving non-depleting induction [23].

In a review from 2015, Malvezzi P. et al., stated that preventing acute rejection after KT can use induction therapy with rabbit anti-thymocyte globulins. Because this induction therapy can have some harmful side effects (e.g., de novo post-transplant cancer or opportunistic CMV infections), it was suggested, as a means to minimize these side effects, to use tacrolimus plus everolimus, and this therapy is efficient in sensitized patients, in recipients from an expanded-criteria donor, and for patients where steroid avoidance is contemplated [24].

Mallat S, et al., are mentioning ganciclovir and valganciclovir as IV antivirals to be used to manage the deleterious outcome of CMV nephritis and monitoring CMV viremia with qPCR assays [25].

In a 5-year study which assessed the relationship between CMV infection and biopsy-proven graft rejection, Dmitrienko S. et al. concluded that, even though current antiviral therapy seems to mitigate the reported harmful effects of CMV infection on biopsy-proven acute rejection (BPAR) or graft survival, BPAR remains a significantly risk factor for both CMV infection and functional graft survival [26].

The first paper identified in PubMed regarding deleterious effects of CMV in KTR was published in 1985, by Bia MJ et al., from Renal Transplant Service, Yale University School of Medicine. The authors concluded that therapy with antithymocyte globulin (ATG) to treat steroid-resistant rejections, has a deleterious influence on the incidence and severity of CMV infection in renal transplant patients, even when the dosage of other immunosuppressive drugs is decreased during ATG therapy [27].  

References: [18-28]: 

18. Koleilat, I.; Kushnir, L.; Gallichio, M.; Conti, D.J. Initiation of a Screening Protocol for Polyoma Virus Results in a Decreased Rate of Opportunistic Non-BK Viral Disease after Renal Transplantation. Transpl Infect Dis 2011, 13, 1–8, doi:10.1111/j.1399-3062.2010.00548.x.

19. Seifert, M.E.; Mannon, R.B.; Nellore, A.; Young, J.; Wiseman, A.C.; Cohen, D.J.; Peddi, V.R.; Brennan, D.C.; Morgan, C.J.; Peri, K.; et al. A Multicenter Prospective Study to Define the Natural History of BK Viral Infections in Kidney Transplantation. Transpl Infect Dis 2024, 26, e14237, doi:10.1111/tid.14237.

20. Zhao, Y.; Zeng, Y.; Lu, R.; Wang, Z.; Zhang, X.; Wu, N.; Zhu, T.; Wang, Y.; Zhang, C. Rapid Point-of-Care Detection of BK Virus in Urine by an HFman Probe-Based Loop-Mediated Isothermal Amplification Assay and a Finger-Driven Microfluidic Chip. PeerJ 2023, 11, e14943, doi:10.7717/peerj.14943.

21. Gouvêa, A.L.F.; Cosendey, R.I.J.; Carvalho, F.R.; Varella, R.B.; de Souza, C.F.; Lopes, P.F.; Silva, A.A.; Rochael, M.C.; de Moraes, H.P.; Lugon, J.R.; et al. Pilot Study of Early Monitoring Using Urinary Screening for BK Polyomavirus as a Strategy for Prevention of BKV Nephropathy in Kidney Transplantation. Transplant Proc 2016, 48, 2310–2314, doi:10.1016/j.transproceed.2016.06.023.

22. Páez-Vega, A.; Gutiérrez-Gutiérrez, B.; Agüera, M.L.; Facundo, C.; Redondo-Pachón, D.; Suñer, M.; López-Oliva, M.O.; Yuste, J.R.; Montejo, M.; Galeano-Álvarez, C.; et al. Immunoguided Discontinuation of Prophylaxis for Cytomegalovirus Disease in Kidney Transplant Recipients Treated With Antithymocyte Globulin: A Randomized Clinical Trial. Clin Infect Dis 2022, 74, 757–765, doi:10.1093/cid/ciab574.

23. Boulay, H.; Oger, E.; Cantarovich, D.; Gatault, P.; Thierry, A.; Le Meur, Y.; Duveau, A.; Vigneau, C.; Lorcy, N. Among CMV-Positive Renal Transplant Patients Receiving Non-T-Cell Depleting Induction, the Absence of CMV Disease Prevention Is a Safe Strategy: A Retrospective Cohort of 372 Patients. Transpl Infect Dis 2021, 23, e13541, doi:10.1111/tid.13541.

24. Malvezzi, P.; Jouve, T.; Rostaing, L. Induction by Anti-Thymocyte Globulins in Kidney Transplantation: A Review of the Literature and Current Usage. J Nephropathol 2015, 4, 110–115, doi:10.12860/jnp.2015.21.

25. Mallat, S.; Moukarzel, M.; Atallah, D.; Abou Arkoub, R.; Mourani, C. Cytomegalovirus Infection Post Kidney Transplant: What Should We Know Now? J Med Liban 2015, 63, 164–169, doi:10.12816/0015841.

26. Dmitrienko, S.; Balshaw, R.; Machnicki, G.; Shapiro, R.J.; Keown, P.A. Probabilistic Modeling of Cytomegalovirus Infection under Consensus Clinical Management Guidelines. Transplantation 2009, 87, 570–577, doi:10.1097/TP.0b013e3181949e09.

27. Bia, M.J.; Andiman, W.; Gaudio, K.; Kliger, A.; Siegel, N.; Smith, D.; Flye, W. Effect of Treatment with Cyclosporine versus Azathioprine on Incidence and Severity of Cytomegalovirus Infection Posttransplantation. Transplantation 1985, 40, 610–614, doi:10.1097/00007890-198512000-00007.    

2. Similarly, their discussion should focus on their results, comparison with data of literature with similar or close methodology, the potential interest for monitoring patients as the gain of expense.

Response to Reviewer: We thank the reviewer for this comment. We have cited more studies and discussed the interrelation between the viruses, relevant clinical outcomes and real-time PCR monitoring, as suggested (lines 346-374). Moreover, we provided some data regarding patient monitoring in Nephrology Department, Dialysis and Renal Transplant Center, “Dr. C.I. Parhon” University Hospital (lines 575-585):  

In Tables 6 and 7 we presented 25 studies which have been published in the last 5 years (2020 – 2024), which used Real Time PCR assays to identify, follow up and guide therapy in CMV / BKV / EBV positive KTRs.

The qPCR equipments and methods were very diverse - cobas® 6800 system, StepOne™ device, AmpliSense CMV-FL test system, Taq-man real-time PCR, establishing a level of 104 CMV DNA copies/mL as the threshold for defining active infection, CMV Re-al-RT Quant kit, real-time PCR method using the CMV Real-TM Quant kit from Sacace, It-aly, using the Argene (bioMérieux) system, nanogen BK virus Q-PCR alert kit, GeneProof BK/JC Virus PCR kit (GeneProof a.s., Czechia), RealStar BKV PCR Kit 1.0 (Altona Diagnos-tics GmbH, Germany), Applied Biosystems 7500 Real-Time PCR System (Applied Biosys-tems, USA), from in house methods, to fully automated systems. This shows the interest of companies for developing these detection and monitoring assays, useful in the case of transplant recipients, but similarly to other medical pathologies (e.g., cervical cancer), these tests lack thorough clinical validation [55, 56]. Each of these qPCR assay should be clinically validated on a significant number of cases and to prove that they get optimal sensitivity, specificity, positive predictive value and negative predictive value, in compar-ison with WHO guideline, for example.

More than this, recent papers are trying to use new modern methods to follow up KTRs who are positive for BKV and CMV. Fernández-Ruiz M et al., from Spain are report-ing the utility of human microRNAs (hsa-miRNAs) as promising biomarkers to identify CMV-seropositive KTRs at risk of CMV reactivation despite detectable CMV-CMI [57]. Sa-linas T et al., from USA, have also reported a urinary cell mRNA profiling for non-invasive diagnosis of acute T cell mediated rejection (TCMR) and BK virus nephropathy (BKVN), which could optimise patient management by minimizing their visits for urine collection [58]. In another study conducted in the USA, Sigdel TK et al. concluded that plasma pro-teomic and transcriptional perturbations impacting humoral and innate immune path-ways are observed during CMV infection and provide biomarkers for CMV disease predic-tion and resolution [59]. All these new and modern methods should also be clinically validated, on large cohorts, in comparison with official international data (e.g., WHO).  

In the practice of Nephrology Department, Dialysis and Renal Transplant Center, “Dr. C.I. Parhon” University Hospital, patients are closely followed up after transplant, regarding the function of the allograft, but also to timely diagnose and try to prevent any kind of complications related to immunosuppression. Patients are tested for viral infec-tions, especially for those classically associated with high risk for transplant recipients (BKV and CMV), according to this center’s protocols, that include also the control of the immunosuppressive medication level that is periodically tested (by checking tacro-linemia, cyclosporinemia etc.). This is in accordance to the previously cited guidelines, in this manner, the medication dose is carefully personalized, to balance the risk of graft re-jection with the risk of infectious complications and viral deleterious effects, and other medication-related side effects.  

3. Comments on the Quality of English Language: It would be wise to check the English by a UK ou US or commonwealth native translator

Response to Reviewer: We thank the reviewer for this comment. We agreed with the Assistant Editor to perform professional language editing by MDPI author services, as suggested.

Round 2

Reviewer 1 Report

Comments and Suggestions for Authors

Dear Authors , thank you very much for this improved version of your work. I do not have any further comments. 

All the best. 

Author Response

Thank you for your thoughtful comments and for recognizing the value of our work. We are deeply honored by your positive feedback.